# Protocol for a cluster-randomised non-inferiority trial of one versus two doses of ivermectin for the control of scabies using a mass drug administration strategy (the RISE study)

Susanna J Lake [1,2] Sophie L Phelan,[3] Daniel Engelman,[1,2] Oliver Sokana,[4] Titus Nasi,[4] Dickson Boara,[4] Christina Gorae,[4] Tibor Schuster,[5] Anneke C Grobler,[6] Millicent H Osti,[1] Ross Andrews,[7] Michael Marks [8,9] Margot J Whitfeld,[10] Lucia Romani [3] John Kaldor,[3] Andrew Steer[1,2]

**Correspondence to**
Dr Andrew Steer;
andrew.steer@rch.org.au

## ABSTRACT

**Introduction** Scabies is a significant contributor to global morbidity, affecting approximately 200 million people at any time. Scabies is endemic in many resource-limited tropical settings. Bacterial skin infection (impetigo) frequently complicates scabies infestation in these settings. Community-wide ivermectin-based mass drug administration (MDA) is an effective control strategy for scabies in island settings, with a single round of MDA reducing population prevalence by around 90%. However, current two-dose regimens present a number of barriers to programmatic MDA implementation. We designed the Regimens of Ivermectin for Scabies Elimination (RISE) trial to investigate whether one-dose MDA may be as effective as two-dose MDA in controlling scabies in high-prevalence settings.

**Methods and analysis** RISE is a cluster-randomised non-inferiority trial. The study will be conducted in 20 isolated villages in Western Province of Solomon Islands where population prevalence of scabies is approximately 20%. Villages will be randomly allocated to receive either one dose or two doses of ivermectin-based MDA in a 1:1 ratio. The primary objective of the study is to determine if ivermectin-based MDA with one dose is as effective as MDA with two doses in reducing the prevalence of scabies after 12 months. Secondary objectives include the effect of ivermectin-based MDA on impetigo prevalence after 12 and 24 months, the prevalence of scabies at 24 months after the intervention, the impact on presentation to health facilities with scabies and impetigo, and the safety of one-dose and two-dose MDA.

**Ethics and dissemination** This trial has been approved by the ethics review committees of the Solomon Islands and the Royal Children's Hospital, Australia. Results will be disseminated in peer-reviewed publications and in meetings with the Solomon Islands Ministry of Health and Medical Services and participating communities.

**Trial registration details** Australian New Zealand Clinical Trials Registry: ACTRN12618001086257. Date registered: 28 June 2018.

## Strengths and limitations of this study

► The cluster-randomised study design follows the implementation of the intervention at a village level
► Follow-up at both 12 and 24 months will demonstrate longer term effects of the intervention
► The sample size (5000 people across 20 villages) achieves a statistical power of 80%
► This study is being conducted in partnership with the Solomon Islands Ministry of Health and Medical Service in a culturally sensitive manner and will build the capacity of local nursing staff
► Scabies prevalence is high in the isolated island villages where this study will be conducted (approximately 20%), therefore results may not be transferable to lower prevalence or urban settings

## BACKGROUND

Scabies is a neglected tropical disease (NTD) caused by infestation with the mite *Sarcoptes scabiei* var. *hominis*. Scabies is a significant contributor to global morbidity, estimated to cause 455 million annual incident cases.[1 2] Transmission occurs primarily as a result of skin-to-skin contact (and rarely due to fomites) and is more common in overcrowded settings, including in many tropical environments where crowding and poverty are prevalent and access to treatment limited.[3 4] The burden of disease is substantial in many Pacific Island Countries where scabies affects one in five people and up to one in two children.[5]

Scabies infestation causes intense itch and discomfort. Furthermore, it is responsible for a considerable proportion of bacterial skin infection (impetigo) in many resource-limited settings.[6–8] Scabies causes a breach

in the skin barrier from scabetic lesions and subsequent scratching, creating an entry point for bacteria, including *Staphylococcus aureus* and *Streptococcus pyogenes*. Resulting impetigo can in turn cause severe infection and immune-mediated disease, including sepsis, glomerulonephritis and possibly rheumatic fever.[9–12]

Treatment guidelines for scabies recommend treatment of the infected individual as well as household contacts.[13 14] Most guidelines recommend treatment with topical acaracides such as permethrin or benzoyl benzoate.[13 14] These medications are effective, if applied to all affected areas for the correct duration, but re-infestation frequently occurs in highly endemic settings where individuals may be exposed to infected household or community members, many of whom may be asymptomatic.[15] Therefore, attention has shifted to simultaneous treatment of whole communities, including those without symptoms of infestation, to reduce prevalence and the rate of transmission.[16] This strategy of mass drug administration (MDA) has been used to successfully control a number of NTDs, including onchocerciasis, lymphatic filariasis (LF), trachoma and soil-transmitted helminths, and there is a growing body of evidence to support MDA for scabies control.[17–22]

Ivermectin is an antiparasitic drug in the avermectin class that is active against the scabies mite. Ivermectin-based MDA for scabies involves offering ivermectin treatment to the whole community, with the exception of young children, pregnant women and others with a contraindication to ivermectin. Permethrin cream is offered as an alternative to ivermectin for these groups. Several studies in Pacific Island Countries with high-prevalence have shown ivermectin-based MDA can reduce the population prevalence of scabies by around 90%.[7 17 23] The Skin Health Intervention Fiji Trial (SHIFT) study in Fiji was the first comparative study to demonstrate the effectiveness of ivermectin MDA for scabies control, finding a reduction in the population prevalence of scabies from 32% at baseline to less than 2% at 12 months.[21] These trials have all used an MDA strategy involving two doses of medication, given 7 to 14 days apart (either to the whole community or those with clinical signs of scabies).[7 23] This is consistent with clinical recommendations for treatment of individuals.[13] Ivermectin is known to lack ovicidal activity, therefore the second dose aims to kill newly hatched mites.[24]

In 2017, the WHO recognised scabies as a NTD, and identified the need for public health action to control scabies in endemic settings.[9] The Strategic and Technical Advisory Group on Neglected Tropical Diseases called for further research into control strategies for scabies and the development of guidelines for the public health use of avermectins.[25]

While ivermectin-based MDA shows great promise as a control strategy for scabies, the requirement for two doses of medication at each MDA round presents barriers to implementation. Drug and implementation costs are doubled compared with single-dose MDA. Distribution is more complex and integration with programmes for other NTDs is difficult. These challenges are amplified in remote island settings where the population is dispersed across difficult to reach villages and funding for programmes is limited. These hurdles may be prohibitive to widespread implementation of scabies control, particularly in low-income settings. Therefore, the optimum dosing strategy for MDA remains an important knowledge gap.[9] For this reason, we designed the Regimens of Ivermectin for Scabies Elimination (RISE) trial.

## METHODS AND ANALYSIS
### Study design
The RISE trial is a prospective, open-label comparison of one dose versus two doses of ivermectin-based MDA for the population-level control of scabies (table 1). Using a cluster-randomised design, 20 villages will be randomised to one of two intervention groups in a 1:1 ratio. Randomisation will occur at the village level, rather than the individual level, as the objective is to determine the dosing regimen for controlling scabies within whole communities. Randomisation minimises the possibility of the anticipated difference in the outcome between each group being confounded. A two-dose regimen is an appropriate comparator (rather than no-treatment or placebo) as two-doses of ivermectin-based MDA is the currently accepted dosing regimen. We chose a non-inferiority design because it is unlikely that a one-dose regimen would be superior in effectiveness to a two-dose regimen.

However, the logistic and pragmatic advantages of a one-dose regimen compared with a two-dose regimen make a non-inferior study appealing.

The prevalence of scabies and impetigo will be measured before the intervention (baseline), and repeated at 12 and 24 months after the intervention. To measure a secondary outcome, the standard Solomon Islands Ministry of Health and Medical Services (MHMS) health facility reporting processes will be used to capture the number of presentations to health facilities with scabies and impetigo in the study catchment area for the 12-month period before the intervention and for the 24-month period after the intervention.

### Aims
The primary objective of this study is to determine if ivermectin-based MDA with one dose is non-inferior to two doses in reducing the prevalence of scabies 12 months after the intervention. The secondary objectives are to assess the impact of ivermectin-based MDA on: population prevalence of scabies after 24 months; population prevalence of impetigo after 12 and 24 months; the number of presentations to health clinics with scabies and impetigo before and after the intervention; the number of adverse events measured by passive surveillance in the 12 months after MDA in each study group. The trial will also be measuring outcomes related to the impact of ivermectin MDA on the prevalence and intensity of

| Table 1 | Key features of the RISE trial |
|---|---|
| Primary objective | To determine if ivermectin-based MDA with one dose is non-inferior to two-doses in reducing prevalence of scabies at 12 months |
| Secondary objectives | Impact of one versus two dose ivermectin-based MDA on:<br>► Population prevalence of scabies at 24 months<br>► Population prevalence of impetigo at 12 and 24 months<br>► Number of presentations to health clinics with scabies and impetigo<br>► Number of adverse events measured by passive surveillance in the 12-month period following MDA |
| Design | Prospective, open-label comparison using a cluster-randomised design |
| Sample size | 20 villages (approximately 5000 participants), randomised in a 1:1 ratio to each intervention group |
| Intervention | Group 1: one dose of ivermectin-based MDA<br>Group 2: two doses of ivermectin-based MDA given 7 to 14 days apart |
| Study setting | Western Province, Solomon Islands (scabies prevalence approximately 20%) |
| Inclusion criteria | All residents in the study villages |
| Exclusion criteria | Participants who meet exclusion criteria will not receive treatment but will still be eligible to enrol in the study and undergo skin examination Exclusion criteria are: allergy to ivermectin or permethrin; treatment within the last 7 days with ivermectin or permethrin; participant declines treatment<br>If ivermectin is contraindicated, topical permethrin will be offered. Contraindications for ivermectin include: pregnancy; breastfeeding an infant less than 7 days old; age less than 2 years; height less than 90 cm; concurrent medication that may interact with ivermectin (for example, warfarin); or severe, acute or chronic illness on the day of MDA |
| Outcome measures | Presence of scabies and impetigo measured by clinical examination<br>Conducted by trained nurses and assessed using the 2020 International Alliance for the Control of Scabies criteria<br>Assessments will be conducted at baseline, 12 and 24 months |

MDA, mass drug administration; RISE, Regimens of Ivermectin for Scabies Elimination.

soil-transmitted helminths; however, this paper focusses on scabies and impetigo outcomes.

## Rationale

This study uses ivermectin-based MDA because the SHIFT trial in Fiji demonstrated the greatest reduction in scabies prevalence after ivermectin-based MDA.[21] Ivermectin is currently the only oral therapy available for scabies and it allows greater compliance than with topical therapy. Oral therapy can be directly observed, ensuring adherence to treatment. Although ivermectin only kills the mature scabies mite and not the eggs, a single dose of treatment simultaneously administered to a whole village may reduce transmission sufficiently to reduce population prevalence. A recent Cochrane review did not find a difference in efficacy of one dose of oral ivermectin compared with two doses of oral ivermectin, but confidence in the effect estimates was low-to-moderate, with poor reporting being a major limitation.[26] A retrospective study in Zanzibar of six rounds of annual single-dose ivermectin MDA for LF showed a 68% to 98% decline in clinical presentations and treatments for scabies, suggesting a one-dose strategy may significantly reduce transmission.[27] By contrast, annual ivermectin MDA for LF did not reduce the prevalence of scabies in Tanzania where the baseline prevalence was less than 5%.[28]

## Study setting and participants

The study will be conducted in Western Province of Solomon Islands (figures 1 and 2). Solomon Islands is a nation in the South Pacific with a population of over 650 000 spread across 900 islands, a geography that presents many challenges for health service delivery.[29] Solomon Islands is classified as a least developed country. It is ranked 152 out of 189 on the Human Development Index.[30] The majority of the population depend on subsistence agriculture in rural locations. We chose Western Province for this study for several reasons: first, there is a high burden of scabies—a 2014 survey estimated an all-age scabies prevalence of 19.2%[6]; second, we expect that there will be relatively little mixing between villages because of the island geography of isolated villages with no road transport; third, there are many villages of appropriate population size (between 180 and 300) for the cluster-randomised design.

Twenty villages will be selected after close consultation with the MHMS. Criteria for selecting the villages include a population of between 180 and 300 people, geographic isolation and willingness to participate in the study. All residents of the 20 selected villages will be eligible to participate. If a resident of a two-dose village does not take the first dose of medication, they will still be eligible to take a dose when the team returns to the village for the second dose.

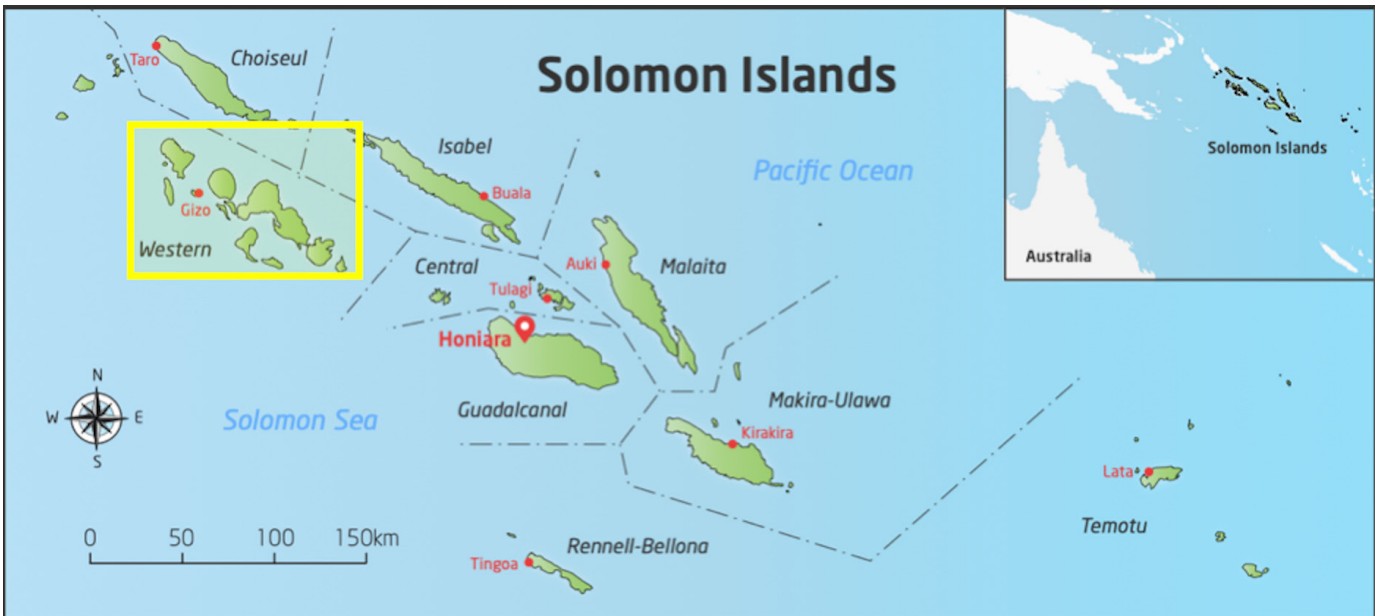

**Figure 1** Study location in Solomon Islands.

This is a community-based study, with analysis conducted at the village level, and therefore, all residents are eligible to participate in the follow-up assessment at 12 and 24 months, regardless of whether they received treatment at baseline. Written informed consent will be obtained from all participants. Participants under the age 18 years will require written consent to be provided by a parent or guardian. Consent will be obtained at each study time point (baseline, 12 months and 24 months) (online supplementary file 1).

### Inclusion and exclusion criteria
#### Inclusion criteria
All participants who provide consent are eligible to participate. If exclusion criteria for treatment are met, then consented participants are still eligible to have their skin examined.

#### Exclusion criteria
Participants who meet any of the following criteria will not receive treatment, but will be eligible to enrol in

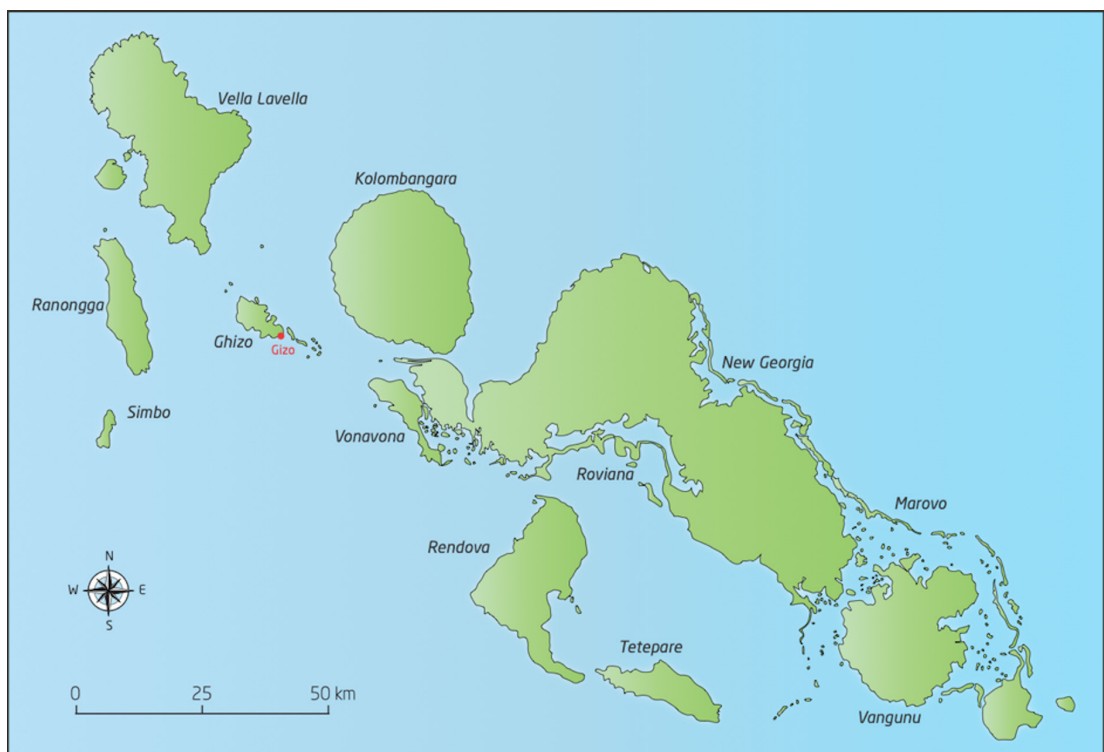

**Figure 2** Study location in Western Province, Solomon Islands.

the study and undergo skin examination: allergy to ivermectin or permethrin; treatment within the last 7 days with ivermectin or permethrin; or declines treatment.

If ivermectin is contraindicated, then topical permethrin will be offered. The contraindications for ivermectin are: pregnancy; breastfeeding an infant less than 7 days old; age less than 2 years; height less than 90 cm; concurrent medication that may interact with ivermectin (eg, warfarin); or severe acute or chronic illness on the day of MDA.

## Patient and public involvement statement

The trial was designed in close consultation with stakeholders at the Solomon Islands MHMS to ensure it was culturally appropriate for the local setting. Staff from the Solomon Islands MHMS contributed to study design and identification of study sites. The study team will comprise a majority of Solomon Islander staff from Western Province. Staff are able to communicate in the local regional languages.

A team of health promotion officers from the Solomon Islands MHMS will conduct community awareness in each village, approximately 1 month prior to MDA. An illustrated information leaflet outlining the study design as well as information about scabies and the treatments will be provided during community visits. A participant information statement that contains contact details for the principal investigator and local investigator will be made available to all village residents. Community awareness and the informed consent process will be conducted in Solomon Islands Pijin and study staff who speak the local regional language will be available to provide further information or clarification as required. Results of the study will be communicated to community leaders and members by the study team.

## Intervention

Oral ivermectin will be offered to all participants, unless there is a contraindication to ivermectin. A dose of 200 µg/kg of ivermectin is recommended for the treatment of individuals with scabies.[13 31] We will aim to dose ivermectin within a range of 150 to 250 µg/kg, as this dose has been effective in previous trials.[21] We will use 6 mg scored tablets. Doses will be rounded to the nearest 3 mg. The tablets will be accurately halved on the score line using a pill cutter as required. As weight scales are generally unsuitable for implementation of MDA, we will use dosing strategies appropriate for larger scale programmatic roll-out.

We will use height-based dosing for children aged less than 15 years, with doses ranging from 3 mg to 12 mg, as is standard for MDA for onchocerciasis and LF.[32 33] Adults will receive a standard dose of 12 mg, with doses adjusted based on visual assessment of body shape (9 mg for adults assessed to be malnourished and 15 mg for adults assessed to be obese). Drug distribution staff will make these assessments based on a series of body shape silhouettes.[34] Dosing of medications for MDA based on

physical appearance has been shown to be accurate and safe.[33] Staff will undergo training and validation for these dosing techniques. Ivermectin will be administered by trained study staff who will directly observe swallowing of the tablets.

Topical permethrin 5% cream will be given to participants meeting exclusion criteria for ivermectin. Permethrin will be dosed according to clinical guidelines.[35] Participants or carers will be counselled to apply the cream to the whole body from the neck down (in infants cream should also be applied to the scalp) and to leave it for 8 to 14 hours, or 4 hours in infants less than 2 months of age.

## Outcome measures

All participants will undergo assessment for symptoms and signs of scabies, impetigo and other skin disease.[36] Assessment will include history questions regarding the presence of itch, contact history and a simplified skin examination. Skin examination will be limited to areas that are usually exposed (arms from above elbow to fingers, legs from above knee to toes, and head and neck). Other areas (including breasts, groin or genitals) will not be examined. Data suggest that, in this setting, a limited examination detects more than 90% cases of scabies.[36] The skin of children less than 2 years age will be examined more generally, as scabies may be more widespread in this age group.[3]

Scabies will be diagnosed according to consensus criteria established by the International Alliance for the Control of Scabies (2020 IACS criteria).[37 38] Categorisation will be based on the identification of typical scabies lesions, typical body distribution of lesions and presence of itch and/or positive contact history. Diagnosis will therefore use levels B (Clinical Scabies) and C (Suspected Scabies) (table 2). Microscopy was not used as it is not feasible or practical for programmatic roll-out in these remote settings. Confirmation of diagnosis with dermatoscopy was not considered feasible as the specialist skills required exceeded the training of the local health workers.

Impetigo will be recorded if papular, pustular or ulcerative lesions surrounded by erythema, or with crusts, pus or bullae are seen.[7] This approach is consistent with the diagnostic processes in previous scabies community intervention trials. Examinations will be conducted by nurses from Western Province. Nurses will receive 1 week of theoretical and practical training in the clinical assessment for scabies and impetigo, including application of the 2020 IACS criteria (online supplementary file 2).[39] Nurses will receive additional training prior to the follow-up surveys at 12 and 24 months.

Other severe skin infections such as ulcers, abscesses or suspected cases of crusted scabies will also be recorded where noted. If these, or other significant medical conditions are noted during the survey, participants will be referred off-study to the local health clinic for assessment and management.

| Table 2 Case definitions for scabies using the 2020 IACS Criteria[38] | | |
|---|---|---|
| **Criteria category** | | **Used in survey** |
| **Confirmed scabies** | | |
| | At least one of the following: | |
| A1 | Mites, eggs or faeces on light microscopy of skin samples | No |
| A2 | Mites, eggs or faeces visualised on an individual using a high-powered imaging device | No |
| A3 | Mite visualised on an individual using dermoscopy | No |
| **Clinical scabies** | | |
| | At least one of the following: | |
| B1 | Scabies burrows | No |
| B2 | Typical lesions affecting male genitalia | No |
| B3 | Typical lesions in a typical distribution and two history features* | Yes |
| **Suspected scabies** | | |
| | At least one of the following: | |
| C1 | Typical lesions in a typical distribution and one history feature* | Yes |
| C2 | Atypical lesions or atypical distribution and two history features* | Yes |

Diagnosis can be made at one of the three levels (A, B or C). A diagnosis of clinical or suspected scabies should only be made if other differential diagnoses are considered less likely than scabies.
*History features include (i) itch and (ii) positive contact history.
IACS, International Alliance for the Control of Scabies.

In addition to skin examination data, we will also collect information on presentations to health facilities in the study villages. Government health facilities in Solomon Islands routinely record the details of all attendances and admissions in paper-based registers. Cases of scabies, local and serious bacterial infections, and other skin diseases are recorded. Facilities report aggregated data using a standardised form each month. Data is transferred electronically through the District Health Information System (DHIS2) to the MHMS Health Information Statistics Unit. We will use the information from DHIS2 to assess the number of presentations to health facilities with scabies and impetigo. Data collected in the 12 months prior to MDA will be compared with data collected in the 24 months following MDA.

### Safety monitoring and reporting
Ivermectin is well tolerated and has a significant dose safety margin with no safety concerns at much higher doses than clinically required (up to 120 mg in adults, approximately 2000 µg/kg).[40 41] Over one billion doses have been distributed for control of onchocerciasis and LF with few effects reported beyond minor, reversible events.[42 43] There have been cases of encephalopathy following ivermectin administration but these have been in the context of loiasis, a disease which has not been detected in Solomon Islands.[44] Ivermectin is on the WHO Model Essential Medicines List and the Solomon Islands Essential Medicines List for the treatment of scabies.[31 45] Although topical benzyl benzoate is the standard treatment of scabies in the Solomon Islands, topical 5% permethrin will be used in the study due to its increased efficacy and lower rate of local side effects.[46] Permethrin is well tolerated with very few side effects, including in infants.[47 48] Nonetheless, we will record all reported adverse events related to treatment using passive monitoring.

Participants will be advised to report any adverse events to clinic nurses or directly to the study team if the adverse event occurs immediately post MDA. The clinic nurses will relay information to the study coordinator who will document the adverse event and send a report to the Principal Investigator who will, in turn, collate adverse events for reporting to the Data Safety Monitoring Board (DSMB). Any serious adverse events or suspected serious adverse reactions will be reported to the study coordinator by the study team, or clinical staff at the clinics and hospitals in the study area. Hospitals will be briefed on the study and provided with comprehensive reporting forms and the MDA schedule. Hospitals will report any admissions or deaths from study villages for 1 month following administration of the first dose of ivermectin. We will retrospectively review mortality records to ensure all deaths from study villages have been captured. We will review routinely collected summary data on all stillbirths from hospitals in the area for 12 months following MDA.

An independent DSMB will provide oversight to the safety and progress of the trial. The DSMB will meet via teleconference prior to the study, in the first 3 months after MDA and at the conclusion of the study. Any serious adverse events and suspected serious adverse reactions will be reported to the DSMB within 7 days.

### Sample size
Sample size calculations were based on scabies prevalence in Western Province of Solomon Islands and the effect size measured in previous studies of ivermectin-based

MDA for scabies.[6 17 21] A standard Monte Carlo simulation method with 1000 repetitions was used to estimate the required sample size and number of villages to achieve statistical power of 80%.[49] We assumed that scabies prevalence across villages would range from 10% to 30% (mean 20%, SD 5%) at baseline.[6] The effect size measured in previous studies with two doses of ivermectin MDA was used to assume the prevalence of scabies 12 months after MDA will be between 3% and 9% (mean 6%, SD 2%) in the one-dose group and between 1% and 5% (mean 3%, SD 1%) in the two-dose group.[17 21] We assumed an average village size of 250 people, with a range of 200 to 300.[50] We considered a non-inferiority margin of 5% (prevalence of scabies at 12 months in the one-dose group minus prevalence at 12 months in the two-dose group) to be relevant from a public health perspective. Based on these assumptions, 20 villages, randomised equally, would be sufficient to achieve the required power.

### Randomisation

An independent statistician will randomise villages to the one-dose or two-dose group in a 1:1 ratio once the 20 study villages have agreed to participate. There will be no stratification within the randomisation process. There will be 10 villages in each group (figure 3).

### Analysis plan

We will account for clustering when calculating all study outcomes by calculating outcomes at the cluster level, and analysing data at the cluster level and not the individual level.[51] The range of cluster-level outcomes will be reported by group.

#### Primary outcome

The prevalence of scabies in each village will be calculated at baseline (0 months) and at 12 months. The prevalence will be calculated by dividing the number of participants with scabies by the denominator (the total number of participants examined for scabies) in each cluster. The denominator will vary at each time point as we will include all participants who consented for skin examination, regardless of their involvement at other time points.

The difference in scabies prevalence between baseline and 12 months will be calculated for each village. The means of these differences will be calculated in the two treatment groups and compared by calculating the difference between the means. If the upper limit of the two-sided 95% CI of the mean difference between the two study groups is less than or equal to 5% (the clinically relevant non-inferiority margin) the one-dose regimen will be considered non-inferior.

#### Secondary outcomes

The analysis for the prevalence of scabies at 24 months and impetigo at 12 and 24 months will be done in the same way as for the primary endpoint.

The change in the number of presentations to health facilities for scabies and impetigo will be analysed in

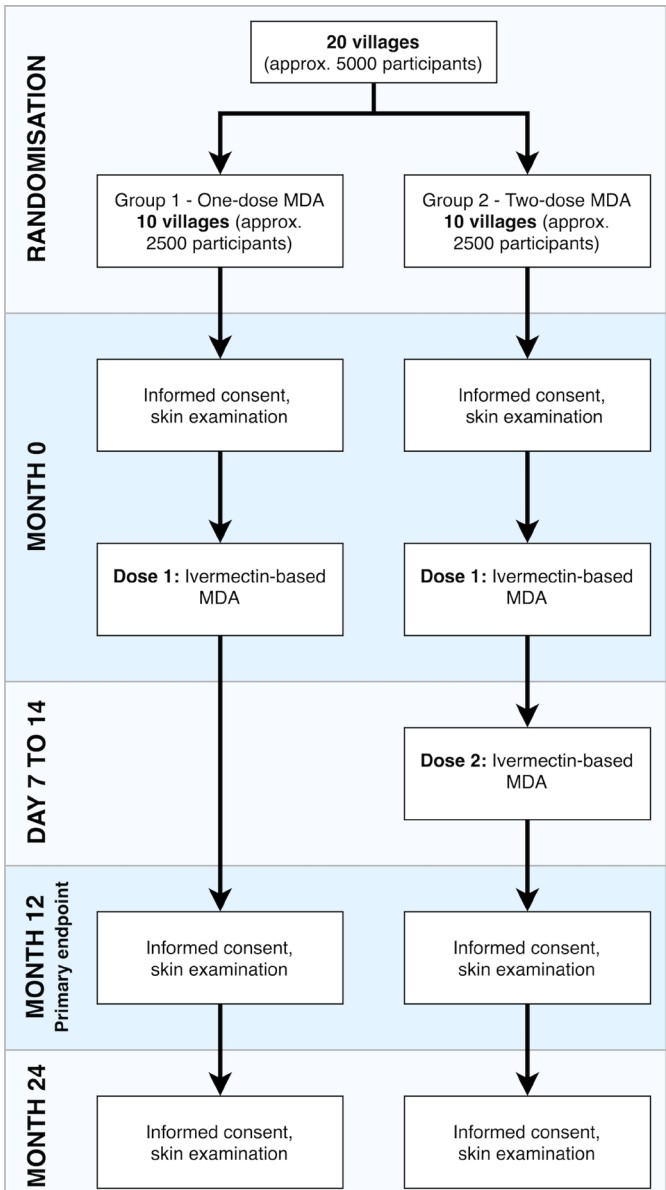

**Figure 3** Study flow diagram. approx., approximately; MDA, mass drug administration

three ways. First, the total number of presentations in the 12 months before MDA will be compared with the number of presentations in months 1 to 12 and 13 to 24 after MDA. Second, we will calculate the proportion of clinic presentations for scabies and impetigo by dividing the number of presentations for scabies and impetigo by the total number of clinic presentations for any condition. We will calculate this proportion for the 12 months before MDA, 1 to 12, and 13 to 24 months after MDA. Calculating the proportion will account for any changes in population size or operational status of health facilities. Third, we will compare the number of clinic presentations for scabies and impetigo in the clinics that service the study villages and compare this with clinic presentation for scabies and impetigo in other clinics in the province, this will be adjusted for population size.

The number of adverse events in each study group will be calculated as a proportion of the total number of participants in each study group that received MDA at baseline. We will also report the number of deaths in the month following MDA in each study group as a proportion of the number of participants in each group.

## Data collection and management
Data will be collected using a combination of paper-based and electronic forms. Paper forms will be stored in locked filing cabinets. Information will be de-identified and participant names will only be recorded on consent forms. Only authorised study staff will be able to access forms. Data will be destroyed after 15 years in compliance with local guidelines. Skin examination data will be collected and managed using REDCap electronic data capture tools hosted at Murdoch Children's Research Institute.[52 53] REDCap is a secure, web-based software platform designed to support data capture for research studies.

## Trial status
Baseline data collection and MDA took place between May and July 2019. A total of 5260 participants were enrolled. Follow-up village data collection is scheduled to take place between May and July 2020 and between May and July 2021.

## Ethics and dissemination
The RISE trial is investigator-initiated and funded by the National Health and Medical Research Council of Australia (GNT1127297). The funder was not involved in protocol development or the study process, including site selection, management, data collection or analysis of the results. The trial is a collaboration between the Murdoch Children's Research Institute, the Solomon Islands MHMS, the Kirby Institute at the University of New South Wales, the London School of Hygiene and Tropical Medicine and the Australian National University.

The trial was designed in accordance with CONSORT (Consolidated Standards of Reporting Trials) guidelines and our reporting of the protocol conforms to the Standard Protocol Items: Recommendations for Interventional Trials (SPIRIT) 2013 checklist.[54–56] This trial has been approved by the Solomon Islands Health Research and Ethics Review Board (HRE005/18) and Royal Children's Hospital Human Research Ethics Committee (38 099A) in Melbourne, Australia, and will be conducted in accordance with Good Clinical Practice.

De-identified data may be made available for further analysis with appropriate approvals. Results of the study will be presented locally and made available to health policy decision makers and clinical staff. Villages that participated in the trial will be presented the results in a culturally appropriate way that is easy to understand and interpret. Participants will have the opportunity for results to be explained to them in their own language

through a series of village meetings as well as printed information leaflets.

## Patientand public involvement
Patientsand/or the public were involved in the design, or conduct, or reporting, ordissemination plans of this research. Refer to the Methods section for furtherdetails.

## DISCUSSION
Scabies is a common disease in many tropical and low-income settings and has been prioritised for control by WHO, but there are still gaps in knowledge to determine the optimum approach to control in settings where scabies is highly endemic.[9] The results of this trial will have an impact on national, regional and global strategies for scabies control.

Island communities in the Pacific have among the highest global prevalence of scabies and understanding how to implement MDA in these settings has the potential for translation into huge public health impact for these communities.[5] However, the results may not be generalisable to populations with a much lower prevalence of the disease, to settings with higher population density or to urban settings. The non-inferiority margin of 5% was determined using available evidence but may not represent the appropriate level of public health significance in all circumstances. A greater or lesser margin may be considered non-inferior in other settings, depending on factors, including baseline disease prevalence, number of rounds planned, costs of implementing each regimen and available resources. This trial is designed to assess a single round of MDA, there is scope for further research to assess the efficacy of repeated annual rounds of MDA. The cluster-randomised design will allow analysis of the impact of MDA at the community level. We will be able to assess the impact of the intervention on the whole community, even for those who will not receive MDA.

If the RISE trial finds that one-dose ivermectin MDA is inferior, then the need for two doses of ivermectin-based MDA would need to be taken into account in decision-making around control strategies for scabies. It would also provide impetus for further research to identify new treatments for scabies that may be able to be implemented with one dose. Approaches may include novel treatments that are ovicidal or other medications with a longer half-life such as moxidectin.[57] If one-dose ivermectin-based MDA is found to be non-inferior to two-dose, then this strategy will be highly attractive for implementation as a public health programme. The lower cost, simplified logistics and ability to integrate with other programmes would make scabies control programmes more feasible in low-income settings.

**Author affiliations**
[1]Tropical Disease Research Group, Murdoch Childrens Research Institute, Parkville, Victoria, Australia
[2]Department of Paediatrics, The University of Melbourne, Melbourne, Victoria, Australia

[3]The Kirby Institute, University of New South Wales, Sydney, New South Wales, Australia
[4]Ministry of Health and Medical Services, Honiara, Solomon Islands
[5]Clinical Epidemiology and Biostatistics Unit, McGill University, Montreal, Quebec, Canada
[6]Clinical Epidemiology and Biostatistics Unit, Murdoch Childrens Research Institute, Parkville, Victoria, Australia
[7]Australian National University, Canberra, Australian Capital Territory, Australia
[8]Clinical Research Department, London School of Hygiene and Tropical Medicine, London, UK
[9]Hospital for Tropical Diseases, London, UK
[10]Department of Dermatology, St Vincent's Hospital, Sydney, New South Wales, Australia

**Acknowledgements** The authors are grateful to the Solomon Islands MHMS and the communities who will participate. We appreciate the support and contributions of Pauline McNeil, Nemia Bainivalu, Gregory Jilini, Michael Larui, Ivan Ghemu, Freda Pitakaka, William Horoto, Jeffrey Korini, Soraya Pina, Yvonne Tuni, Selina Maena, Frederick Neqo, Susana Vaz Nery and Naomi Clarke. The RISE (Regimens of Ivermectin for Scabies Elimination) study team includes: Sana Bisili, Aisling Byrne, Sharmillah Jack, Arthur Keremama, Alam Khatak, Erica Lazu, Relinta Manaka, Davis Pesala, Deanne Seppy, Winter Sino, Patson Solomon, Stephen Tiazi and Salote Wickham. This study is dedicated to Dr Tenneth Dalipanda, former Permanent Secretary of the Solomon Islands Ministry of Health and Medical Services. Dr Tenneth was committed to improving the health of Solomon Islanders. He was an advocate for public health research and without his support, we would not have been able to conduct this research and many studies before it.

**Contributors** Study concept and design were conducted by the investigators: AS, DE, JK, LR, MJW, MM, RA, TS, OS and TN. Critical revision of concept and design, and intellectual input in the study protocol was done by all authors: SJL, SLP, DE, OS, TN, DB, CG, TS, ACG, MHO, RA, MM, MJW, LR, JK and AS. Drafting of the protocol was done by SLP and ACS, with review by all authors. Drafting of the manuscript was done by SJL. Critical revision of the manuscript was performed by all authors. Study supervision is conducted by the investigators.

**Funding** This research is funded by the National Health and Medical Research Council of Australia (NHMRC) (APP1127297). DE, LR, JK and ACS are supported by fellowships from the NHMRC. MM is supported by the UK National Insitute for Health Research.

**Map disclaimer** The depiction of boundaries on this map does not imply the expression of any opinion whatsoever on the part of British Medical Journal (or any member of its group) concerning the legal status of any country, territory, jurisdiction or area or of its authorities. This map is provided without any warranty of any kind, either express or implied.

**Competing interests** None declared.

**Patient consent for publication** Not required.

**Provenance and peer review** Not commissioned; externally peer-reviewed.

**ORCID iDs**
Susanna J Lake http://orcid.org/0000-0002-5508-5430
Michael Marks http://orcid.org/0000-0002-7585-4743
Lucia Romani http://orcid.org/0000-0001-9038-5300

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
