## [Reviewer comments · BMJ Open]

ARTICLE DETAILS

TITLE (PROVISIONAL)	Protocol for a cluster randomised non-inferiority trial of one versus two doses of ivermectin for the control of scabies using a mass drug administration strategy (the RISE study)
AUTHORS	Lake, Susanna; Phelan, Sophie; Engelman, Daniel; Sokana, Oliver; Nasi, Titus; Boara, Dickson; Gorae, Christina; Schuster, Tibor; Grobler, Anneke; Osti, Millicent; Andrews, Ross; Marks, Michael; Whitfeld, Margot; Romani, Lucia; Kaldor, John; Steer, Andrew

VERSION 1 – REVIEW

REVIEWER	Lorenz von Seidlein Mahidol-Oxford Tropical Medicine Research Unit (MORU) Faculty of Tropical Medicine Mahidol University 420/6 Rajvithi Road Bangkok 10400 Thailand
REVIEW RETURNED	17-Feb-2020

GENERAL COMMENTS	This is a review of the study protocol for a cluster randomised non-inferiority trial of one versus two doses of ivermectin for the control of scabies using a mass drug administration strategy (the RISE study). As my instructions indicate “For studies that are ongoing, it is generally the case that very few changes can be made to the methodology. As such, requests for revisions are generally clarifications for the rationale or details relating to the methods. If there is a major flaw in the study that would prevent a sound interpretation of the data, we would expect the study protocol to be rejected.” I therefore restrict my review to my general impression and the minor suggestions which could help the reader to understand the manuscript. Overall this is a well written manuscript. As so often the in protocols the introduction provides a comprehensive, helpful overview of the topic with up-to-date data, including a summary of use of avermectins in scabies control, the need for mass drug administrations, and the reasons for and against a 2-dose strategy. The study design a 2-arm, cluster randomised non-inferiority trial is perfectly appropriate to answer the research question. My minor suggestions would be to state the dates of study start and study end? It is conventionally accepted that protocols include a table indicating the procedures (y-axis) and timeline (x-axis). Adding such a table could be helpful?
---

REVIEWER	Corinna Dressler
-----------------	------------------

	Charité Universitätsmedizin Berlin
REVIEW RETURNED	28-Feb-2020

GENERAL COMMENTS	Clear and very good description of the processes and procedures. I have three remarks/questions: 1) Diagnostic criteria are described but the actual outcome(s) in terms of scabies is not quite specified. Past trials used "cured" or "clear/almost clear" , which were defined very differently.. Could you please clarify your definition of clear/cure/scabies free? 2) Statistical Analysis: Cluster trial appropriate analysis is mentioned but no details. Do you plan to do any subgroup analysis or can you pre-specify covariates ? 3) The trials is open label. Would it be possible to do a blinded outcome assessment or perhaps a blinded statistical analysis to reduce bias?
--

REVIEWER	Jo Middleton Department of Primary Care and Public Health; NIHR Global Health Unit on Neglected Tropical Diseases, Brighton and Sussex Medical School, UK I am a member of the International Alliance for the Control of Scabies (IACS), a relatively small network of which many of the authors are members. Few work on scabies compared to other NTDs, so most potential reviewers of this protocol would likely also be IACS members. I do not consider this a competing interest (I am in no way involved in the work outlined in the protocol), but thought I should mention this for transparency.
REVIEW RETURNED	04-Mar-2020

GENERAL COMMENTS	The protocol is well written and clear, and concerns a valuable study for one of the most neglected of the neglected tropical diseases. I hope the following suggestions can improve the manuscript, and I am happy to review the paper again if the editors feel necessary. I am unable to give some page numbers, as these are not showing on most of the PDF. Line 12-13: "transmission occurs as a result of skin to skin contact and is more common in overcrowded settings". This paragraph gives a general background to scabies, and as such I think it is worth mentioning that transmission can also happen as a result of fomites. I would not expect this to change the design of the intervention (I myself am doing scabies work elsewhere in the tropics and am not considering environmental decontamination), however in such an introductory paragraph I would suggest something along the following lines: 'transmission occurs primarily as a result of skin to skin contact (and also sometimes via fomites [reference Mellanby 'Scabies in 1976: https://doi.org/10.1177%2F146642407709700112]), and is more common in overcrowded...' P8 lines 3-10, and later 35-38: Both sections state that having to give two doses of medication presents barriers to implementation, integration with other health programmes, and that the drug implementation costs double. I agree with all this, however if this was read by somebody who isn't involved in Mass Drug Administrations (MDAs) in relatively remote settings it might be unclear why this is. Given this is such a key part of the justification
---

	for this trial I suggest that elaborating with some practical examples would aid the reader. Section on scabies diagnosis methodology, paragraph starting: "scabies will be diagnosed according to". The paragraph includes the following "confirmation of diagnosis with microscopy and dermatoscopy is not feasible in this remote setting". Whilst standard lab microscopy may be unfeasible in this setting, it is not clear to me why dermatoscopy would automatically be. I am not a proponent of introducing such technologies into clinical practice in otherwise medically underserved communities, as they are very often too expensive to be maintained by in-country health services. However, no reasons have been given by the authors as to why using dermatoscopes (lightweight, rechargeable, hand-held devices) was unfeasible in remote settings as part of their research project. As this study has already collected (some) data I do not mention this with the expectation that the study design would be altered, simply that sufficient justification is given in the protocol. Same paragraph as above: "nurses will receive one week of theoretical and practical training in the clinical assessment for scabies, impetigo, application of the IACS diagnostic criteria. Nurses will receive additional training prior to the follow-up surveys of 12 and 24 months". This is excellent. I would suggest a very basic syllabus for this training could be included in appendix so it can be taken up by other teams doing similar work around scabies MDAs. Once again this is not in any way to doubt the validity of the content of the training, more to aid reproducibility of the methodology by other teams. "Villages that participated in the trial will be presented the results in a culturally appropriate way that is easy to understand and interpret." This explanation is not sufficient for either evaluation or replication. I suggest stating exactly what community dissemination is planned. Village meetings? Written briefings? Primarily picture-based educational leaflets or posters? Also when it is expected the community dissemination will be carried out should be stated (i.e. how long after data collection). I note (apologies if I've missed anything) that the only place where any limitations of the study are mentioned is in the article summary: "scabies prevalence is high in the isolated villages where the study will be conducted (approximately 20%), therefore results may not be transferable to low prevalence or urban settings." This limitation should be mentioned in the actual text (maybe in the discussion section which is relatively brief). If the authors are aware of other limitations of their study they should also be stated in the main text. All my points are relatively minor, though I feel worth addressing, and I congratulate the authors for such a well-designed study. I very much look forward to reading what the study finds, which will be of high importance for those of us involved in scabies MDAs.
--	--

VERSION 1 – AUTHOR RESPONSE

Reviewer 1:

1. To state the dates of study start and study end?

As above, the date for Mass Drug Administration, 12 month and 24 month data collection are under 'Trial Status' in the Methods section: "Baseline data collection and MDA took place between May and July 2019. A total of 5,260 participants were enrolled. Follow-up village data collection is scheduled to take place between May and July 2020 and between May and July 2021."

2. It is conventionally accepted that protocols include a table indicating the procedures (y-axis) and timeline (x-axis). Adding such a table could be helpful?

Figure 3 is a flowchart which provides an overview of the procedures and timeline. Please advise if this is not sufficient and if a table is preferred.

Reviewer 2:

1. Diagnostic criteria are described but the actual outcome(s) in terms of scabies is not quite specified. Past trials used "cured" or "clear/almost clear", which were defined very differently. Could you please clarify your definition of clear/cure/scabies free?

This trial does not aim to investigate if individual participants are cured of scabies. Rather, we assess the prevalence of scabies at a community level at baseline and 12 (and 24) months after the intervention. Therefore, we used the international diagnostic criteria developed by IACS (Engelman et al, BJD, 2020) at both timepoints.

2. Statistical Analysis: Cluster trial appropriate analysis is mentioned but no details. Do you plan to do any subgroup analysis or can you pre-specify covariates ?

We do not plan on doing any subgroup analysis as the intervention is unlikely to be tailored to specific subgroups and the study is not large enough to be adequately powered for subgroup analyses. We will adjust for any covariates.

3. The trials is open label. Would it be possible to do a blinded outcome assessment or perhaps a blinded statistical analysis to reduce bias?

It is not possible to do a blinded outcome assessment because the nurses doing the skin examinations are aware which villages received one and two doses of ivermectin. We cannot use different nurses at 12 and 24 months for skin examinations because this would lead to greater sources of bias. We will take every precaution to make sure bias is not introduced during data analysis but as statistical analysis is done by a researcher and the data is unblinded it is not possible to do blinded statistical analysis. Statistical analysis will be done on the whole data set together using Stata. Data will not be divided into each village or group for data analysis. There are strict pre-defined diagnostic criteria for scabies and impetigo. A positive or negative scabies diagnosis cannot be changed at statistical analysis. The analysis method and Stata syntax are determined before analysis.

Reviewer 3:

1. I think it is worth mentioning that transmission can also happen as a result of fomites. I would suggest something along the following lines: 'transmission occurs primarily as a result of skin to skin contact (and also sometimes via fomites [reference Mellanby 'Scabies in 1976:

<https://doi.org/10.1177%2F146642407709700112>]), and is more common in overcrowded...'

We have altered the above sentence in 'Background' to: "Transmission occurs primarily as a result of skin-to-skin contact (and rarely due to fomites) and is more common in overcrowded settings, including in many tropical environments where crowding and poverty are prevalent and access to treatment limited."

2. P8 lines 3-10, and later 35-38: Both sections state that having to give two doses of medication presents barriers to implementation, integration with other health programmes, and that the drug implementation costs double. I agree with all this, however if this was read by somebody who isn't involved in Mass Drug Administrations (MDAs) in relatively remote settings it might be unclear why this is. Given this is such a key part of the justification for this trial I suggest that elaborating with some

practical examples would aid the reader.

We have added to 'Background': "These challenges are amplified in remote island settings where the population is dispersed across difficult to reach villages and funding for programs is limited."

3. Section on scabies diagnosis methodology, paragraph starting: "scabies will be diagnosed according to". The paragraph includes the following "confirmation of diagnosis with microscopy and dermatoscopy is not feasible in this remote setting". Whilst standard lab microscopy may be unfeasible in this setting, it is not clear to me why dermatoscopy would automatically be. I am not a proponent of introducing such technologies into clinical practice in otherwise medically underserved communities, as they are very often too expensive to be maintained by in-country health services. However, no reasons have been given by the authors as to why using dermatoscopes (lightweight, rechargeable, hand-held devices) was unfeasible in remote settings as part of their research project. As this study has already collected (some) data I do not mention this with the expectation that the study design would be altered, simply that sufficient justification is given in the protocol. We agree that low-cost dermatoscopy devices may have a roll in some field settings, but require specialised skills and training. We have added the following clarification: "Microscopy was not used as it is not feasible or practical for programmatic roll-out in these remote settings. Confirmation of diagnosis with dermatoscopy was not considered feasible as the specialist skills required exceeded the training of the local health workers".

4. Same paragraph as above: "nurses will receive one week of theoretical and practical training in the clinical assessment for scabies, impetigo, application of the IACS diagnostic criteria. Nurses will receive additional training prior to the follow-up surveys of 12 and 24 months". This is excellent. I would suggest a very basic syllabus for this training could be included in appendix so it can be taken up by other teams doing similar work around scabies MDAs. Once again this is not in any way to doubt the validity of the content of the training, more to aid reproducibility of the methodology by other teams.

We have added an overview of the training to the supplementary.

5. "Villages that participated in the trial will be presented the results in a culturally appropriate way that is easy to understand and interpret." This explanation is not sufficient for either evaluation or replication. I suggest stating exactly what community dissemination is planned. Village meetings? Written briefings? Primarily picture-based educational leaflets or posters? Also when it is expected the community dissemination will be carried out should be stated (i.e. how long after data collection). We have altered the sentence to better describe the village feedback process: "Participants will have the opportunity for results to be explained to them in their own language through a series of village meetings as well as printed information leaflets."

6. I note (apologies if I've missed anything) that the only place where any limitations of the study are mentioned is in the article summary: "scabies prevalence is high in the isolated villages where the study will be conducted (approximately 20%), therefore results may not be transferable to low prevalence or urban settings." This limitation should be mentioned in the actual text (maybe in the discussion section which is relatively brief). If the authors are aware of other limitations of their study they should also be stated in the main text.

The second paragraph of the discussion currently has the sentence "However, the results may not be generalisable to populations with a much lower prevalence of the disease, to settings with higher population density, or to urban settings." We have also added: "The non-inferiority margin of 5% was determined using available evidence but may not represent the appropriate level of public health significance in all circumstances. A greater or lesser margin may be considered non-inferior in other settings, depending on factors including baseline disease prevalence, number of rounds planned, costs of implementing each regimen and available resources."

VERSION 2 – REVIEW

REVIEWER	Lorenz von Seidlein Mahidol-Oxford Tropical Medicine Research Unit (MORU) Faculty of Tropical Medicine Mahidol University 420/6 Rajvithi Road Bangkok 10400 Thailand
REVIEW RETURNED	24-May-2020

GENERAL COMMENTS	I have been asked to review the protocol of an ongoing trial. The publication of the protocol should help with the interpretation of trial results to be published in the future. The journal makes the following recommendation for the review of protocols. The dates of the study should be included in the manuscript. The authors have done this. If there is a major flaw in the study that would prevent a sound interpretation of the data, the protocol should be rejected. The study addresses a very reasonable research question. The methodology proposed in the protocol is appropriate to address the question. The protocol is well written and should be acceptable for publication. I have two minor suggestion. 1) the authors could consider adding a summary table of the most relevant features of the protocol for a rapid overview. 2) the inclusion/exclusion criteria as currently presented could confuse readers quickly scrolling through the protocol. It may be helpful to add the contraindications for ivermectin administration to the exclusion criteria or refer to the contraindications in the exclusion criteria so that the reader understands that young children will be exclude from the trial.
--

REVIEWER	Jo Middleton Brighton and Sussex Medical School; University of Sussex
REVIEW RETURNED	15-May-2020

GENERAL COMMENTS	The authors have addressed my previous comments, and where necessary added detail or made changes. I wish the authors well with their important study and look forward to reading its findings.
---

VERSION 2 – AUTHOR RESPONSE

Reviewer 1:

1. The authors could consider adding a summary table of the most relevant features of the protocol for a rapid overview.

A summary table of the key features of the trial including primary objective, secondary objectives, design, sample size, intervention, study setting, inclusion criteria, exclusion criteria and outcome measures has been included at the end of the Methods and Analysis section.

2. The inclusion/exclusion criteria as currently presented could confuse readers quickly scrolling through the protocol. It may be helpful to add the contraindications for ivermectin administration to the exclusion criteria or refer to the contraindications in the exclusion criteria so that the reader understands that young children will be excluded from the trial.

We have moved the contraindications to ivermectin to the 'exclusion criteria' section from the

'intervention'. Young children will not be excluded from the trial if they are less than two years of age or less than 90 cm height they will receive topical permethrin.

The exclusion criteria section now reads:

“Exclusion criteria. Participants who meet any of the following criteria will not receive treatment, but will be eligible to enrol in the study and undergo skin examination: allergy to ivermectin or permethrin; treatment within the last 7 days with ivermectin or permethrin; declines treatment.

If ivermectin is contraindicated, then topical permethrin will be offered. The contraindications for ivermectin are: pregnancy; breastfeeding an infant less than seven days old; age less than two years; height less than 90 cm; concurrent medication that may interact with ivermectin (for example, warfarin); or severe acute or chronic illness on the day of MDA.”

Note that the inclusion and exclusion criteria have also been included in the summary table.

VERSION 3 – REVIEW

REVIEWER	Lorenz von Seidlein MORU, Thailand
REVIEW RETURNED	21-Jul-2020
GENERAL COMMENTS	I have reviewed the track changes in the revised manuscript and found my queries adequately addressed.